# On the Species Identification of Two Non-Native Tilapia Species, Including the First Record of a Feral Population of *Oreochromis aureus* (Steindachner, 1864) in South Korea

**DOI:** 10.3390/ani13081351

**Published:** 2023-04-14

**Authors:** Ju Hyoun Wang, Hee-kyu Choi, Hyuk Je Lee, Hwang Goo Lee

**Affiliations:** 1Aquatic Ecosystem Research Laboratory, Department of Biological Science, College of Science & Engineering, Sangji University, Wonju 26339, Republic of Korea; 2Molecular Ecology and Evolution Laboratory, Department of Biological Science, College of Science & Engineering, Sangji University, Wonju 26339, Republic of Korea

**Keywords:** first record in Korea, invasive species of tilapia, morphological identification, molecular identification

## Abstract

**Simple Summary:**

Tilapia is farmed in many parts of the world. Tilapia species have likely been introduced into new habitats, altering these habitats at times detrimentally. In this study, the morphological and molecular identification of the tilapia species inhabiting the Dalseo Stream in South Korea was undertaken. Currently, it is reported that only one species of *O. niloticus* inhabits natural rivers in Korea. However, except for the aquaculture population, the *O. aureus* natural population, whose habitat has not been reported in natural rivers, was identified for the first time in this study. The results of this study provide useful data for establishing management plans for invasive species management and the information necessary for identifying tilapia species that exist in Korea.

**Abstract:**

Tilapia is an invasive species that has become widely distributed around the world. In Korea, introduced tilapia into its aquatic ecosystem for the first time with a species from Thailand in 1955, and later additionally introduced two more species from Japan and Taiwan, thus securing a total of three species of tilapia (*O. niloticus*, *O. mossambicus* and *O. aureus*) as food resources. Since then, *O. niloticus* has been reported to inhabit certain streams with thermal effluent outlets. Morphological species identification is very difficult for tilapia and a combined analysis of morphological and molecular-based species identification is therefore necessary. This study investigated a tilapia population that inhabits a thermal effluent stream (Dalseo Stream) in Daegu Metropolitan City, Korea, in order to conduct a morphological and genetic species identification of this population. In total, 37 tilapia individuals were sampled. The results of the morphological and genetic species identification analyses found that two species, *O. aureus* and *O. niloticus*, inhabit the Dalseo Stream. In Korea, the habitat of the *O. niloticus* natural population has been reported, but the *O. aureus* natural population has not been reported. Thus, we observed for the first time that a new invasive species, *O. aureus*, inhabits a stream in Korea. They are known to cause disturbances to aquatic organisms (e.g., fish, aquatic insects, plankton, aquatic plants) and the habitat environment (e.g., water quality, bottom structure). Accordingly, it is important to study the ecological effects of *O. aureus* and *O. niloticus* on the corresponding freshwater ecosystem closely and to prepare a management plan to prevent the spread of these species, as they are notoriously invasive.

## 1. Introduction

Tilapia, invariably known as *Oreochromis*, *Sarotherodon* and *Tilapia*, commonly refers to three genera in the family Cichlidae [1], with approximately 100 species worldwide [2]. Tilapia of the genus *Oreochromis* is now farmed in many countries (e.g., China, Indonesia, Tanzania, Uganda, Brazil, Taiwan, Vietnam, Philippines, Egypt), with its global production reaching 5.6 million tons in 2020 [3]. *Oreochromis niloticus* (Linnaeus, 1758), *Oreochromis mossambicus* (Peters, 1852) and *Oreochromis aureus* (Steindachner, 1864) are the most commonly farmed species of tilapia in the world [4,5]. In order to secure food resources, Korea introduced *O. aureus*, *O. niloticus* and *O. mossambicus* from Thailand, Japan, and Taiwan, respectively [6].

Tilapia species are among the most widely distributed invasive species in the world and have been introduced into many aquatic ecosystems through various routes or methods (e.g., release, fish farm escape) [7,8]. Invasive species can negatively affect the functioning of freshwater ecosystems by changing the trophic levels and food chain of native species [9,10]. When tilapia is introduced outside of its native range, it can be detrimental to native species through destructive spawning, prey competition and habitat competition [11,12,13]. Tilapia is also known to have increasingly altered the water quality and to have decreased biodiversity [14,15].

In general, tilapia fish do not survive at water temperatures below 10 °C, and some hybrid types are known to not survive at temperatures of approximately 7 °C to 8 °C [16,17]. In Korea, the water temperature in winter falls below 10 °C, meaning that the tilapia population may not survive in most Korean natural freshwater ecosystems during winter. However, the occurrence of *O. niloticus* was observed for the first time in the Hwangguji Stream, through which thermal effluent flows, in 2008 [18]. Since then, *O. niloticus* has been continuously reported in rivers affected by thermal effluent outlets (e.g., Dalseo Stream, Geumho River, Gokgyo Stream) [19,20]. The source of the inflow of tilapia to these rivers and streams is unclear, but it is likely that they escaped from fish farms or are ornamental fish that were released [11,13,21,22].

It is challenging to undertake morphological species identification of the different species of tilapia that have been introduced and that inhabit many areas of the world [2,23]. In particular, different tilapia species are almost impossible to distinguish through morphological methods given the existence of many hybrids [2,24]. Most tilapia species farmed today are based on hybrid combinations of *O. niloticus*, *O. aureus* and *O. mossambicus* [25], and species identification requires both morphological and genetic analyses. As one example, species classification was conducted through the morphological and genetic analyses of a tilapia population living in the Philippines [23]. The morphological analysis classified these into two species, *O. niloticus* and *O. mossambicus*, but the genetic analysis confirmed *O. aureus* as well, and it was also reported that hybrid specimens were identified among these three species [23]. It is therefore necessary to conduct both a morphological and a genetic analysis to identify and distinguish tilapia of the genus *Oreochromis*, from many hybrids have arisen [22,23,26]. On the other hand, Kim and Park (1990) [6] conducted a karyotype analysis of farmed tilapia introduced in Korea and confirmed the presence of three species of *O. niloticus*, *O. mossambicus*, and *O. aureus*. Previously, because the habitat of the natural population had not been reported, species identification was conducted only for the farmed tilapia population. However, all natural populations reported from 2008 to the present have been identified based on morphological methods [18,19,20], and no studies have identified natural populations through genetic methods. Therefore, in order to clearly identify this species of tilapia, which is difficult to distinguish morphologically, morphological and genetic identification studies are needed for natural populations living in Korea.

In this study, we investigate an invasive tilapia population in the Dalseo Stream in Korea and conduct a morphological-based species identification. In addition, using the cytochrome oxidase (COI) gene, we perform molecular-based phylogenetic species identification using mtDNA haplotypes registered in the GenBank database. Our study will provide information on invasive tilapia occurring in Korea through morphological and molecular species identification, and also providing information that will aid in the management of invasive tilapia.

## 2. Materials and Methods

### 2.1. Sample Collection

We collected 37 tilapia (except for young fishes) individuals using cast nets (7 × 7 mm) from three sites (St. 1: N 35°52′57.07″, E 128°33′16.51″; St. 2: N 35°53′00.35″, E 128°32′23.86″; St. 3: N 35°53′08.26″, E 128°32′00.53″) within the Dalseo Stream in South Korea on 20 May 2020 and 10 November 2021 (Figure 1). The collected tilapia individuals were photographed at the site. A small piece (approximately 3–5 mm) of fin tissue was collected from each individual and preserved immediately in 99% ethanol and stored at 4 °C until the genetic analysis [27]. The capture and use of the animals for this study were approved by the institutional ethics committee of Sangji University in Korea. However, no research permits were required for the collection of fish samples as this species is not listed as endangered in Korea.

### 2.2. Classification according to the Morphological Identification

Tilapia individuals collected in the field were classified according to the morphological characteristics established by [2]. The morphology classification was divided into the number of dorsal spines and rays, tail fin patterns, tail fin colors, nuptial coloration, and gill rakers. In addition, the analysis involved a Kruskal–Wallis test to confirm whether the difference between the number of dorsal spines and rays and the number of gill rakers, which are classification keys that can be used to identify tilapia species, was statistically significant. Statistical tests were carried out using the IBM SPSS Statistics 21 software.

### 2.3. Identification

The specific morphological characteristics of *Oreochromis aureus* are as follows [2,28,29]. The caudal fin and dorsal fin edge of blue tilapia are a broad bright red or vermilion. The head of the male fish changes to a bright metallic shade of blue during the breeding season, and he also displays color on the edge of his dorsal fin and on the margin of his caudal fin. A breeding female fish develops a pale orange color on the edges of her dorsal and caudal fins. The caudal fin edge has no distinct bands or stripes. Dorsal spines number 15 to 16, total dorsal spines and rays number 27 to 30, and gill rakers number 18 to 26 (Table 1).

The specific morphological characteristics of *Oreochromis niloticus* are as follows [2,30,31]. The gill rakers are short, the maxilla and lower jaw are equal, the pectoral fin is pointed, and the dorsal, pectoral and anal fins are blunt. The head and trunk of breeding males are suffused with red, and in some localities the lower jaw, pelvis and anterior part of the anal fin are black. The dorsal fin margin is dark gray or black and the caudal fin shows distinct, regular dark stripes. Dorsal spines number 16 to 18, total dorsal spines and rays number 29 to 31, and gill rakers number 27 to 33 (Table 1).

The specific morphological characteristics of *Oreochromis mossambicus* are as follows [2,32]. These tilapia have fine pharyngeal teeth, breeding males are black (not in some cultured strains) with white lower parts on the head, and they have red dorsal and caudal fin margins. The remnants of a striped and barred pattern are often visible in females, juveniles and non-breeding males, as a series of mid-lateral and dorsal blotches. The jaws of adult males are greatly enlarged, and they have a concave dorsal head profile. The male genital papilla is simple or slightly notched, and the caudal fin is not densely scaled. Dorsal spines number 15 to 18, total dorsal spines and rays number 25 to 31, and gill rakers number 14 to 20 (Table 1).

### 2.4. Mitochondrial DNA Sequencing

Genomic DNA was extracted from the dorsal fin using a P&C animal genomic DNA extraction kit (Biosolution, Korea). For phylogenetic species identification, mtDNA COI (612bp) was used as a molecular marker. The forward primer (FF2d_FW: 5-TTCTCCACCAACCACAARGAYATYGG-3) and the reverse primer (FR1d_RV: 5-CACCTCAGGGTGTCCGAARAAYCARAA-3) used were sourced from a previous study [33].

Polymerase chain reaction (PCR) amplification was performed with 9.9 μL of sterilized distilled water, 1.5 μL of 10 × Green Buffer (Thermo Scientific Inc., Waltham, MA, USA), 0.5 μM of the forward/reverse primer, 1.5 μL of 2.0 mM dNTPs (Bio Basic Inc., Markham, Ontario, Canada), 0.1 U of DreamTaq DNA polymerase (Thermo Scientific Inc.) and 1 μL of genomic DNA (5–15 ng μL^−1^) (total 15 μL) in a 2720 thermal cycler (Applied Biosystems, Waltham, MA, USA). The following thermal cycling conditions were used: initial denaturation at 94 °C for 2 min followed by 35 cycles of denaturation at 94 °C for 30 s, annealing at 52 °C for 40 s and an extension at 72 °C for 1 min, followed by a final extension at 72 °C for 10 min. The PCR products were checked on 2% agarose gels stained with RedSafe (iNtRON Biotechnology, Daejeon, Korea). The amplified PCR products were purified enzymatically with Exonuclease I (New England BioLabs Inc., Ipswich, MA, USA) and shrimp alkaline phosphatase (New England BioLabs). The purified mtDNA fragments were subjected to direct sequencing in the forward and reverse directions using the same forward and reverse primers used in the PCR case with a BigDye Terminator 3.1 Cycle Sequencing Ready Reaction Kit (Applied Biosystems) in an ABI 3730xl automated DNA sequencer (Applied Biosystems).

### 2.5. Genetic Data Analysis

The DNA sequences were edited using CHROMAS ver. 2.1.1 software and aligned using BIOEDIT ver. 7.2.5 [34] software. The COI gene sequences obtained from each of the 37 samples and from the 12 foreign sequences registered in the National Center for Biotechnology Information (NCBI) were used for molecular-based species identification (Table 2). A phylogenetic analysis was conducted using the neighbor joining (NJ) method with MEGA ver. 7.0 [35] and using 1000 bootstrap trials. Additionally, for the between-species and genetic distance analyses, the Kimura-2-parameter (K2P) distance was calculated during the phylogenetic tree analysis after 1000 bootstrap trials with the Kimura model using the MEGA X program [36].

To track the origin of tilapia inhabiting the Dalseo Stream, we tried to compare it with the COI genetic data of all the *O. aureus* and *O. niloticus* registered in the NCBI. A total of 330 individuals of *O. aureus* and *O. niloticus* were confirmed, and among them, individuals with a base sequence length shorter than 631 bp were excluded for use. Therefore, 17 individuals of *O. aureus* registered in NCBI, 184 individuals of *O. niloticus*, 1 individual of *Tilapia zillii* as an out-group, and 2 haplotypes (haplotypes 1, 2; GenBank accession No. OQ402198, OQ402199) identified in the Dalseo Stream were analyzed in the same manner as above (Appendix A).

## 3. Results

### 3.1. Photographic Record

From the Dalseo Stream in Daegu, South Korea (N 35°53′00.35″, E 128°32′23.86″), the specimens were collected by JH Wang on 20 May 2020 and 10 November 2021; 12 specimens of *Oreochromis aureus* and 25 specimens of *Oreochromis niloticus* were collected and photographed (Figure 2).

### 3.2. Morphological Species Identification

The results of the morphological analyses showed that the collected tilapia specimens were identified as two species, *O. aureus* and *O. niloticus*, and no individuals showing the phenotype of *O. mosambicus* were identified (Table 3 and Table 4). In most of the collected tilapia, species could not be distinguished via a meristic analysis of the dorsal fins. However, by counting the gill rakers during the dissection step, *O. aureus* and *O. niloticus* were clearly distinguished. The statistical differences between the species in terms of their number of dorsal spines and their number of gill rakers were analyzed; and it was found that only the number of gill rakers showed statistically significant differences between the two species (*p* < 0.001) (Figure 3).

### 3.3. Molecular Identification

As a result of the genetic analyses, two species, *O. aureus* and *O. niloticus*, were identified. In total, the 37 tilapia specimens analyzed from the Dalseo Stream population contained only two haplotypes (Table 4). The genetic distance (K2P distance) between the two haplotypes identified was 8.0%. Four main groups were produced in the NJ tree, which were determined based on the reference sequences clustered to each group (Figure 4). These groups were designated as *O. niloticus* and *O. mossambicus* (Group A), *O. niloticus* (Group B), *O. aureus* (Group C), and *T. zillii* (Group D). All three *Oreochromis* genus groups were clustered away from the *T. zillii* group. Among the three *Oreochromis* genus groups, Groups A and B were clustered closer together with high bootstrap support. Group A included *O. niloticus*, *O. niloticus* × *O. mossambicus*, *O. niloticus* GIFT and *O. mossambicus* × *O. niloticus*, and four clustered subgroups; haplotype 1 formed one clade with *O. niloticus* in the species from the Philippines (bootstrap value = 100). *O. niloticus* in the species from the Philippines in group A was found to be relatively distant from the *O. niloticus* (GenBank accession No. GU370126) of group B, which is known as the original African population. Haplotype 2 was confirmed to form one clade with the African original population, *O. aureus* (GenBank accession No. GU370125) (bootstrap value = 100).

### 3.4. Presumption of the Tilapia Introduction Area

A NJ tree analysis was conducted using two haplotypes (haplotypes 1 and 2) along with the sequence data of 201 individuals of *O. aureus* and *O. niloticus* that were available from the GenBank database; this was in order to trace the origin of the *O. aureus* and *O. niloticus* identified in the Dalseo Stream (Appendix A and Figure 5). Haplotype 2 (*O. aureus*) formed a clade with the blue tilapia found in several countries (e.g., Philippines, Nigeria, the Republic of the Congo, the Democratic Republic of the Congo, Guinea, Mexico, Guatemala), meaning that the original population cannot be identified. However, haplotype 1 (*O. niloticus*) formed a clade with *O. niloticus* (GenBank accession Nos. HQ654742, KT307766, KT307765), which is found only in the Philippines. Therefore, it is conceivable that the *O. niloticus* population living in the Dalseo Stream originated from this region.

## 4. Discussion

### 4.1. Comparison of Morphological Analyses Results

We undertook a morphological species identification of the non-native species of tilapia that populate the Dalseo Stream, into which thermal effluent outlets flow. As a result of the morphological classification of tilapia individuals collected from the Dalseo Stream, these specimens could be classified into two species: Blue Tilapia (*O. aureus*) and Nile Tilapia (*O. niloticus*). The morphological and meristic characteristic features that distinguish the two species include the dorsal fin, the tail fin patterns and colors, the spawning season nuptial coloration, and the gill rakers [2,31]. Individuals of a relatively small size (total length) or with damaged caudal fins could not be used for species identification because their nuptial coloration and morphological traits were ambiguous. With regard to the number of dorsal fins, it was found that there was no difference between the mean values of the two species (*p* > 0.05). According to previous studies, although there are some differences in the species depending on their habitats and environments, it is generally known that *O. aureus* has 18–26 gill rakers [2,41] and that *O. niloticus* has 27–33 [2]. The analysis of the gill rakers in the 37 samples of tilapia showed that the average number for *O. aureus* was 23–26 (24.0 ± 1.11), while that for *O. niloticus* was 28–31 (29.4 ± 1.0), figures that are consistent with those found in earlier research. There was also a clear difference in the number of gill rakers between the two species (*p* < 0.001). Therefore, in order to distinguish the two species morphologically, identification using gill rakers is considered to be the most effective method.

### 4.2. Comparison of Molecular Analyses Results

As a result of the genetic species identification analysis of the 37 samples of *Oreochromis* spp., the genetic distance between the two haplotypes was found to be 8%. Previous studies have shown that genetic distances of more than 2–3% in animals may indicate different species [42], and there is a significant genetic difference between the two hap-lotypes found here. The NJ tree analysis determined that the two species were *O. aureus* and *O. niloticus*, but Maranan et al. 2016 [23] found that the aquaculture and natural population of tilapia in the Philippines is mostly a hybrid species of *O. niloticus*, *O. mossambicus* and *O. aureus*. In addition, there is evidence of extensively introgressed *O. niloticus* and *O. mossambicus* in the Philippines [43]. Therefore, the *O. niloticus* (GenBank accession Nos. KT307766, KT307765) registered with the NCBI may also be a hybrid. According to the results of this study, haplotype 1 was relatively distant genetically from *O. niloticus* (GenBank accession No. GU370126), known as the African native population, and was found to have the nearest genetic distance to a Chinese population (GenBank accession No. DQ426668), known to be a hybrid species (*O. niloticus* × *O. mossambicus* hybrid). Therefore, haplotype 1 may be a hybrid of *O. niloticus* and *O. mossambicus*, and further research using additional samples with more genetic markers (e.g., nuclear markers, recombination-activating genes RAG1 and RAG2) should be conducted to identify such hybrids or species [44].

### 4.3. Potential Impact of the Aquatic Ecosystem

We confirmed for the first time that the wild population of *O. aureus*, an invasive and non-native species that is different from the previously reported *O. niloticus* natural population, inhabits a stream in Korea. Currently, there are several native species of fish (e.g., *Squalidus gracilis majimae*, *Squalidus chankaensis tsuchigae*, *Odontobutis platycephala*, *Odontobutis interrupta*) living in Dalseo Stream. Tilapia is known to have a negative impact on the existing native fish and their surrounding habitats (e.g., water plants, water quality, turbidity) when it enters a new habitat [45,46]. In particular, as tilapia is very able to adapt to various habitat environments, it is very important to protect and manage the aquatic environment properly when introducing these species [47]. The *Oreochromis aureus* and *O. niloticus* species that inhabit the Dalseo Stream in Korea are considered to be highly invasive by the ISSG (Invasive Species Specialist Group), and are considered to be and are managed as one of the world’s top 100 disturbing species [41,48]. Many countries, including Brazil, Nicaragua, and South Africa, recognize the seriousness of the habitat disturbances caused by these species and prepare management measures to prevent them from spreading (e.g., physical blocking, population control through direct capture, biological control using potential predators) [22,49,50,51]. The negative impact that tilapia have in Korea ecosystem may not have detected due to a lack of data, because most of them live only in limited environments with thermal effluent outlets. Hence, it is important to study the effects that the tilapia that inhabit streams in Korea have on indigenous species and river ecosystems, and it seems evident that systematic measures are needed to manage the invasive tilapia species.

## 5. Conclusions

As a result of the morphological and genetic analyses of tilapia populations living in the Dalseo Stream conducted in this study, *O. niloticus* were identified, and *O. aureus*, an unrecorded species in Korea, was confirmed for the first time. (ICZN code: urn:lsid:zoobank.org:pub:B6EFDD2F-3284-4037-A5D3-ACC52942A593) In a morphological analysis, it was confirmed that the number of gill rakers was 100% consistent with the results of a genetic species analysis. Therefore, in order to distinguish the two species morphologically, classification using gill rakers is considered to be the most effective method. With regard to *O. niloticus*, it was found that it formed a clade with an *O. niloticus* population that only inhabits the Philippines; therefore, it is presumed that this regional population was introduced and lives in the area of the Dalseo Stream. On the other hand, since the tilapia of the *O. niloticus* species from the Philippines is reported to be a hybrid population, the *O. niloticus* in the Dalseo Stream is also likely to be a hybrid. Research thus far has confirmed that both species can live only in a limited environment due to the decreased water temperatures in winter. However, hybrid *O. niloticus* can survive at relatively low water temperatures compared to original native *O. niloticus*, and is likely to cause biological disturbance due to its high ability to adapt to various habitats. Therefore, a risk assessment of tilapia should be conducted through the mid-to long-term monitoring of rivers inhabited by tilapia populations. In addition, it is judged that it is necessary to prepare physical (e.g., physical blocking, population control through direct capture) and institutional management measures to prevent the spread of tilapia.

## Figures and Tables

**Figure 1 animals-13-01351-f001:**
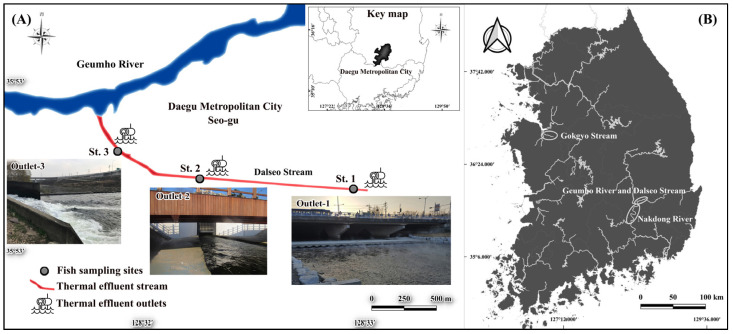
(**A**) The map of sampling sites in the Dalseo Stream and the Geumho River, Daegu Metropolitan City, South Korea. (**B**) Distribution of tilapia habitats identified in Korea.

**Figure 2 animals-13-01351-f002:**
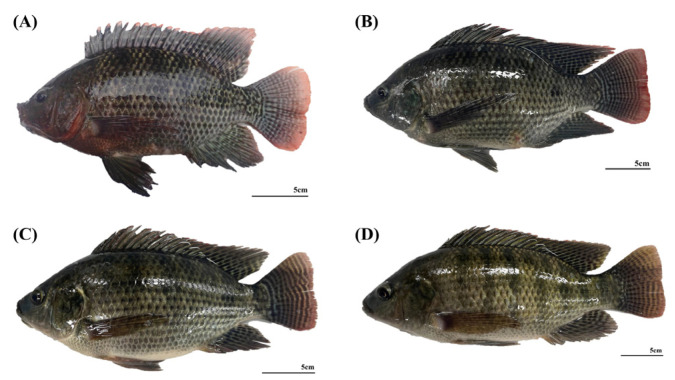
(**A**) *Oreochromis aureus* (male), (**B**) *Oreochromis aureus* (female), (**C**) *Oreochromis niloticus* (male) and (**D**) *Oreochromis niloticus* (female) from the Dalseo Stream, South Korea (photographed by JH Wang).

**Figure 3 animals-13-01351-f003:**
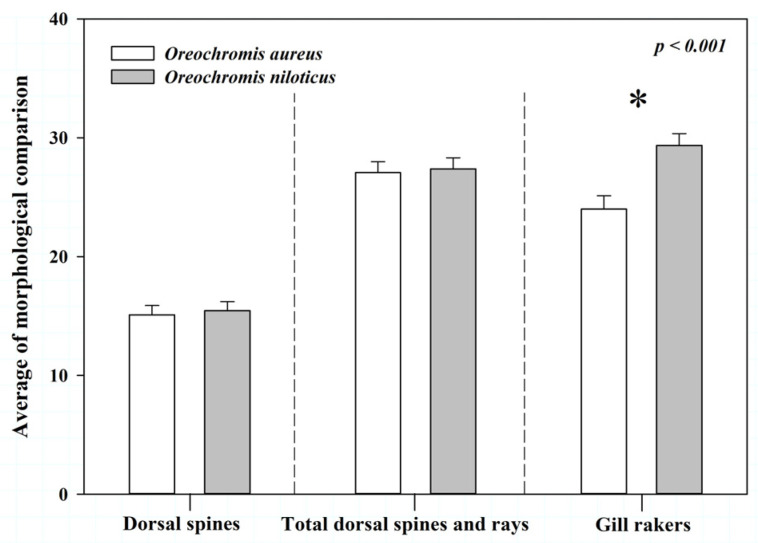
Comparisons of average values of the number of dorsal spines, total number of dorsal spines and rays, and number of gill rakers between *O. aureus* and *O. niloticus*. Each box represents the mean of the morphological comparison (white box is *O. aureus*, *n* = 12; gray box is *O. niloticus*, *n* = 25; Mann–Whitney U tests: * *p* < 0.001).

**Figure 4 animals-13-01351-f004:**
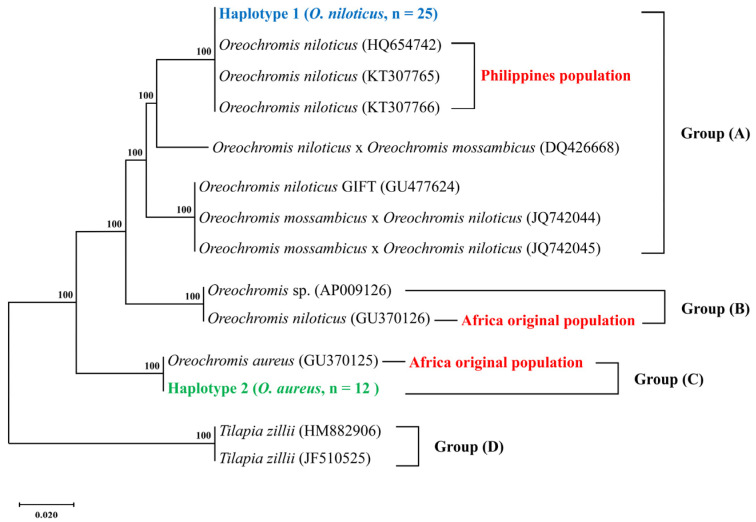
Neighbor-Joining (NJ) tree of 49 COI sequences from *Oreochromis* species, *Oreochromis* hybrids, and *Tilapia zillii* (outgroup taxon) computed via the Kimura 2-parameter method (Kimura, 1980). Bootstrap supports using 1000 replicates are shown. Sequences obtained from GenBank display their corresponding accession numbers. The two haplotypes shown (haplotype 1: *n* = 25, haplotype 2: *n* = 12) represent the 37 COI sequences of genus *Oreochromis* specimens collected in the study, with specimens exhibiting identical sequences grouped under the same haplotypes. These were designated as Group A (*O. niloticus* and *O. mossambicus*), Group B (*O. niloticus*), Group C (*O. aureus*), and Group D (*T. zillii*).

**Figure 5 animals-13-01351-f005:**
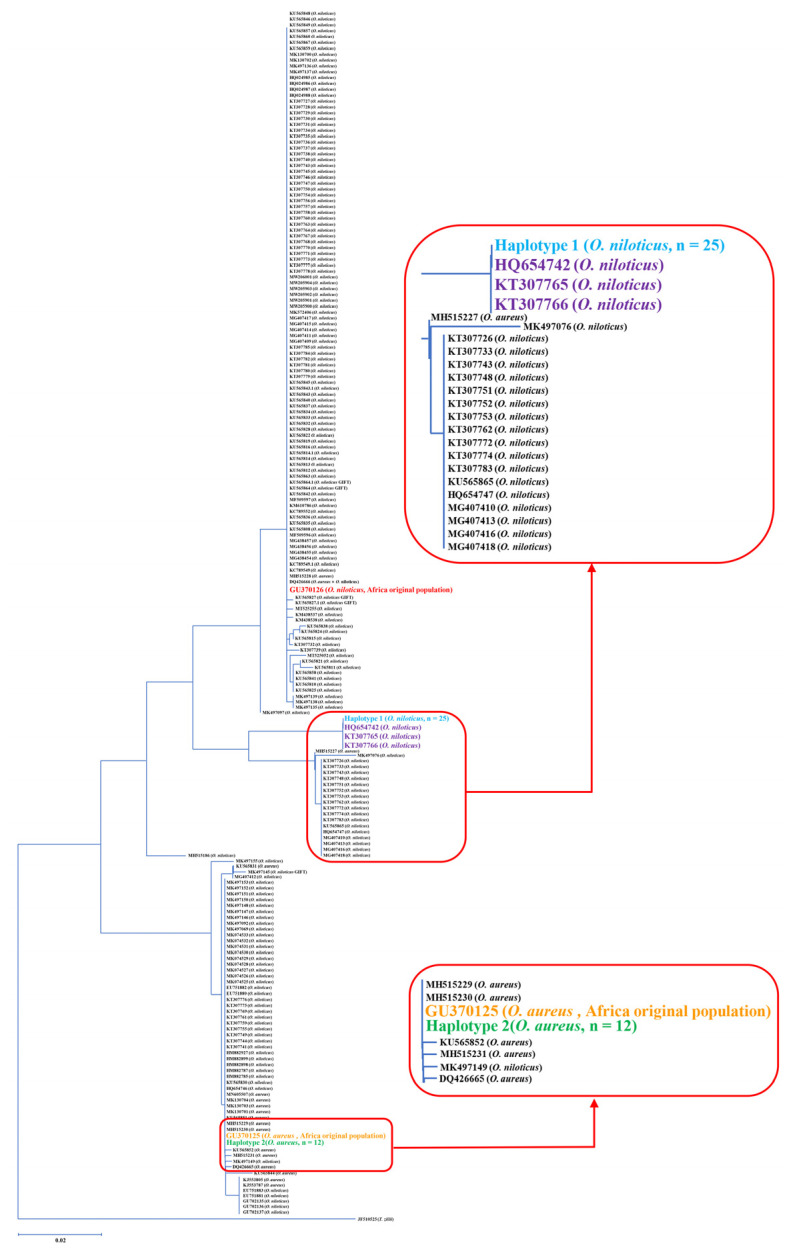
Neighbor-Joining (NJ) tree of 204 COI sequences from *Oreochromis niloticus*, *Oreochromis aureus*, and *Tilapia zillii* (outgroup taxon) computed via the Kimura 2-parameter method (Kimura, 1980). Bootstrap supports using 1000 replicates are shown. Sequences obtained from GenBank display their corresponding accession numbers. The two haplotypes shown (haplotype 1, haplotype 2) represent the COI sequences of the genus *Oreochromis* specimens collected in the study. The yellow haplotype refers to the original population (*O. aureus*), the red haplotype refers to the original population (*O. niloticus*), and the purple haplotype refers to the *O. niloticus* population, which only inhabit the Philippines.

**Table 1 animals-13-01351-t001:** Morphological and meristic comparisons between *O. aureus, O. niloticus* and *O. mossambicus*, according to reference [2,28,29,30,31,32].

Morphological Comparison	Blue Tilapia (*O. aureus*)	Nile Tilapia (*O. niloticus*)	Mozambique Tilapia (*O. mossambicus*)
Dorsal spines	15–16	16–18	15–18
Total dorsal spines and rays	27–30	29–31	10–13
Gill rakers	18–26	27–33	14–20
Banding on caudal fin	No distinct bands/stripes	Distinct, regular dark stripes	No distinct bands
Male breeding coloration	Metallic blue	Red	Black
Dorsal fin margin	Vermilion	Dark grey or black	Red

**Table 2 animals-13-01351-t002:** List of additional sequences downloaded from GenBank for phylogenetic analysis for species identification.

Species	GenBank Accession Number	Reference	Sampling Area
*Oreochromis aureus*	GU370125	[37]	Africa
*Oreochromis niloticus*	HQ654742	[38]	Philippines
	KT307765	[23]	Philippines
	KT307766		
	GU370126	[37]	Africa
*O. niloticus × O. mossambicus*	DQ426668	unpublished	China
*O. mossambicus × O. niloticus*	JQ742044	unpublished	unknown
	JQ742045		
*Oreochromis niloticus* GIFT	GU477624	unpublished	unknown
*Oreochromis* sp.	AP009126	[39]	unknown
*Tilapia zillii*	HM882906	[40]	Nigeria
	JF510525		

**Table 3 animals-13-01351-t003:** Morphological and meristic comparisons of *O. aureus* and *O. niloticus* from the Dalseo Stream.

Morphological Comparison	Blue Tilapia (*O. aureus*)	Nile Tilapia (*O. niloticus*)
Dorsal spines	14–16 (average 15.1 ± 0.8)	15–17 (average 15.4 ± 0.8)
Total dorsal spines and rays	26–28 (average 27.1 ± 0.9)	26–29 (average 27.4 ± 0.9)
Gill rakers	23–26 (average 24.0 ± 1.1)	28–31 (average 29.4 ± 1.0)
Banding on caudal fin	No distinct bands/stripes	Distinct, regular dark stripes
Dorsal fin margin	Vermilion	Dark grey or black

**Table 4 animals-13-01351-t004:** Species identification of each specimen based on morphological and genetic analyses.

Species Designation	Morphological Identification	Frequency of Each Haplotype Per Specimen Designation		No. of Samples	Genetic Identification/GenBank Accession Number
		Haplotype 1	Haplotype 2		
*O. aureus*	3, 7, 8, 9, 11, 16, 17, 33, 34, 35, 36, 37	-	3, 7, 8, 9, 11, 16, 17, 33, 34, 35, 36, 37	12	*O. aureus*/GU370125
*O. niloticus*	1, 2, 4, 5, 6, 10, 12, 13, 14, 15, 18, 19, 20, 21, 22, 23, 25, 25, 26, 27, 28, 29, 30, 31, 32	1, 2, 4, 5, 6, 10, 12, 13, 14, 15, 18, 19, 20, 21, 22, 23, 25, 25, 26, 27, 28, 29, 30, 31, 32	-	25	*O. niloticus*/HQ654742, KT307765, KT307766

## Data Availability

The mtDNA COI sequences of two haplotypes obtained for this study have been deposited in GenBank under the accession numbers OQ402198-OQ402199.

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
