# Peer review of "On the Species Identification of Two Non-Native Tilapia Species, Including the First Record of a Feral Population of Oreochromis aureus (Steindachner, 1864) in South Korea"

_animals, 2023, doi:10.3390/ani13081351_

Round 1

Reviewer 1 Report

Review manuscript: “On the species identification of two non-native tilapia species 2 including the first record of a feral population of Oreochromis aureus (Steindachner, 1864) in South Korea “. This manuscript studies the invasive species management of two non-native tilapia species in South Korea. Since Invasive species can negatively affect freshwater ecosystems by changing trophic levels, and the food chain, and lead to the introduction of new pathogens to native species.

As such, I find the manuscript to be an important scientific contribution.

I suggest only minor modifications:

Abstract

1.     The first time a new species name appears in the manuscript – should be in a full name as you did in your title - Oreochromis aureus (Steindachner, 1864)

Introduction

2.     Replace “generally” with “general” (line 54).

3.     Since different tilapia species are hard to distinguish through morphological methods (lines 66-67), it’s also important to write the reasons why it’s important.

4.      Please consider adding a short paragraph on the management of invasive tilapia species.

Materials and Methods

5.     Please provide a map with a better resolution in Fig 1.

Results

6.     Very good section

Discussion

7.     Consider adding a paragraph regarding the management. It should be adding in the impacts of the aquatic ecosystem section. Without strong management formation/tools for the management of invasive tilapia, it will be limited to distinguishing tools for tilapia species only.

Conclusion

1.     Consider moving more data to the discussion section. The conclusion section is too long.

Author Response

Response to Reviewer 1 Comments

Review manuscript: “On the species identification of two non-native tilapia species 2 including the first record of a feral population of Oreochromis aureus (Steindachner, 1864) in South Korea “. This manuscript studies the invasive species management of two non-native tilapia species in South Korea. Since Invasive species can negatively affect freshwater ecosystems by changing trophic levels, and the food chain, and lead to the introduction of new pathogens to native species.

As such, I find the manuscript to be an important scientific contribution.

Thank you for the helpful comments for improving the manuscript.

I suggest only minor modifications:

Point 1. The first time a new species name appears in the manuscript – should be in a full name as you did in your title - Oreochromis aureus (Steindachner, 1864)

Response 1: The full name of the species is stated at the beginning of the introduction. (Line 43-44)

Point 2. Replace “generally” with “general” (line 54)

Response 2: Revised. (Line 56)

Point 3. Since different tilapia species are hard to distinguish through morphological methods (lines 66-67), it’s also important to write the reasons why it’s important.

Response 3: In the introduction, it is stated that it is important to perform genetic analysis together for tilapia species identification because it is difficult to distinguish tilapia morphologically.

Point 4. Please consider adding a short paragraph on the management of invasive tilapia species.

Response 4: Adding a paragraph about species management in the introduction was not considered because it was written in the Review and Conclusion section.

Point 5. Please provide a map with a better resolution in Fig 1.

Response 5: Revised. (Line 112)

Point 6. Very good section

Response 6: We thank the reviewer for the positive comments.

Point 7. Consider adding a paragraph regarding the management. It should be adding in the impacts of the aquatic ecosystem section. Without strong management formation/tools for the management of invasive tilapia, it will be limited to distinguishing tools for tilapia species only.

Response 7: The impact of both species on aquatic ecosystems is mentioned in the discussion (4.3. Potential impact of the aquatic ecosystem). It also describes the management of these species.

Point 8. Consider moving more data to the discussion section. The conclusion section is too long.

Response 8: Revised. (Line 326-348)

Author Response

Response to Reviewer 2 Comments

Thank you for the helpful comments for improving the manuscript.

We substantially revised a previous version of our manuscript by considering all the comments and issues raised by the reviewer.

Reviewer 3 Report

Ju Hyoun Wang et al applied a morphological approach and molecular mtDNA barcoding system to identify the non-native tilapia species inhabiting a thermal effluent stream close to Daegu city in South Korea. Based on these analyses, Oreochromis niloticus was confirmed to be present as documented previously and Oreochromis aureus, the first report of this tilapia species in this country.

The species identification of tilapia species is difficult given overlapping morphological traits and introgression.  Key to this is the inclusion and comparison of pure populations of tilapia species with a known history and an absence of hybridisation to the test (unknown) population. These reference groups of known origin, pure populations need to be incorporated into the study to enable a clear comparison and species association, and distinguish hybrid individuals. Include O. mossambicus as it is mentioned as being farmed in the country.  The identification of these fish as O. niloticus and O. aureus is ‘likely’ i.e. rather than definitive when the closest affiliations are to probable impure individuals, which suggests an admixture.  Wording of the text needs to be revised to account for this likelihood of the study fish being introgressed and suspected multiple species contribution.

I have the following recommendations and queries to clarify the content;

Oreochromis in full first mention (Line 13, 16, 21), shorten elsewhere (Line 113)

Replace ‘types’ with more specific term e.g. species? (Line 21)

Do you have data of the tilapia fish farm locations in South Korea can this be summarised in relation to the water courses sampled and likely introduction in the environment? (line 62)

Confusing section. Re-word and detail the methods used in speciation of previous studies (Line 76 – 82)

Justify the number (37) of fish sampled, was this sufficient (representative of the sampling area) and comparable to other similar studies? (Line 94)

A map of the sampling locations of previous studies where O. niloticus was identified within South Korea next to the current project sampling location in Figure 1 would be of interest. (Line 103)

Reference Kruskal-Wallis test (Line 109)

Provide enzyme concentrations only volume is stated (Line 132)

Genomic (Line 134)

State specific Taq enzyme (Line 145)

Round up to nearest ng (Line 146)

Include species of fish with confirmed pure populations and reference (Line 169)

Which sequences were used? (Line 175)

In the Table label and separate your results by column and the reference data (Line 200)

Include reference values of each pure species attributes alongside the study fish Figure 3 (Line 202)

Include more examples of pure populations Figure 4 for O. aureus and O. niloticus (native to Africa and Middle East) to distinguish from hybrids e.g. bold font for pure lineages, plain font hybrids. Include fish with known sampling location (line 222)

F3 instead of 3 etch, clearer that number relates to Fish Sample Number Table 4. Simplify Table no need to repeat numbers in Haplotype (Line 229)

Colour code country origin (coloured circle) and highlight pure populations. Expand tree for study samples of O. aureus as well as O. niloticus. It is unclear the origin of the closest individual fish matching Haplotype 2 (Line 243)

State ranges of earlier work (Line 264)

Discuss why not O. mossambicus. Can this species be discounted by morphology? (Line 267)

The two species were ‘most closely associated with’ (Line 273). Speciation is not definitive as affiliations are to mixed populations of fish. Include fish for comparison where degree of hybridisation and purity is already known. 

Philippines likely hybrid lineage, provide evidence of relative species contribution and origin. What happens with morphological features in hybrids? Discuss likely % species contribution (Line 276)

Reference and detail appropriate methods available eg SNPs. Include these characterised populations in the comparison (Line 283)

List native fish present.  Is there evidence of competition since the introduction of tilapia into the country? (Line 287)

Supplementary Materials

Table S1 No need to provide sequence. State accession number, origin, whether pure or suspected introgressed and add Reference for comparative samples or state project sequences as ‘Current study’.

Author Response

Response to Reviewer 3 Comments

Ju Hyoun Wang et al applied a morphological approach and molecular mtDNA barcoding system to identify the non-native tilapia species inhabiting a thermal effluent stream close to Daegu city in South Korea. Based on these analyses, Oreochromis niloticus was confirmed to be present as documented previously and Oreochromis aureus, the first report of this tilapia species in this country.

The species identification of tilapia species is difficult given overlapping morphological traits and introgression. Key to this is the inclusion and comparison of pure populations of tilapia species with a known history and an absence of hybridisation to the test (unknown) population. These reference groups of known origin, pure populations need to be incorporated into the study to enable a clear comparison and species association, and distinguish hybrid individuals. Include O. mossambicus as it is mentioned as being farmed in the country. The identification of these fish as O. niloticus and O. aureus is ‘likely’ i.e. rather than definitive when the closest affiliations are to probable impure individuals, which suggests an admixture. Wording of the text needs to be revised to account for this likelihood of the study fish being introgressed and suspected multiple species contribution.

Thank you for the helpful comments for improving the manuscript.

In the case of O. mossambicus, it is necessary to check whether it is currently farmed in Korea. Recent records show that only O. niloticus is farmed in breeding grounds and fishing grounds.

Therefore, it is difficult to include O. mossambicus in the results of this study.

Point 1: I have the following recommendations and queries to clarify the content; Oreochromis in full first mention (Line 13, 16, 21), shorten elsewhere (Line 113)

Response 1: Since the text starts from the introduction, the entire scientific name was presented in the introduction, shorten elsewhere

Point 2: Replace ‘types’ with more specific term e.g. species? (Line 21)

Response 2: Revised. We revised the manuscript according to the minimum word number of the Article type.

Point 3: Do you have data of the tilapia fish farm locations in South Korea can this be summarised in relation to the water courses sampled and likely introduction in the environment? (Line 62)

Response 3: There is no record of the place used as a tilapia farm in the past, so it is impossible to check. Since the places currently used as fish farms are far from the river, it is judged that it is impossible to enter through the escape of fish farms. However, tilapia discharged from fishing sites can be introduced into the river, and possible to introduce artificially by artificial release.

Point 4: Confusing section. Re-word and detail the methods used in speciation of previous studies (Line 76 – 82)

Response 4: Revised. (Line 78-86)

Point 5: Justify the number (37) of fish sampled, was this sufficient (representative of the sampling area) and comparable to other similar studies? (Line 94)

Response 5: The study was conducted on adults except for young fishes. Although there are no clearly defined number of samples and sampling areas, sufficient objects have been obtained for statistical analysis and comparison. (Line 100)

Point 6: A map of the sampling locations of previous studies where O. niloticus was identified within South Korea next to the current project sampling location in Figure 1 would be of interest. (Line 103)

Response 6: Revised. (Line 112)

Point 7: Reference Kruskal-Wallis (Line 109)

Response 7: Revised. (Line 123-124)

Point 8: Provide enzyme concentrations only volume is stated (Line 132)

Response 8: We deleted details due to the limited number of words.

Point 9: Genomic (Line 134)

Response 9: Thank you for your finding our mistake.

Point 10: State specific Taq enzyme (Line 145)

Response 10: Revised. (Line 160)

Point 11: Round up to nearest ng (Line 146)

Response 11: Revised. (Line 160-161)

Point 12: Include species of fish with confirmed pure populations and reference (Line 169)

Response 12: The data on pure population are presented in the text and in Table S1.

Point 13: Which sequences were used? (Line 175)

Response 13: The COI gene was used, and the individual sequences used are shown in Table S1.

Point 14: In the Table label and separate your results by column and the reference data (Line 200)

Response 14: Table 1 is the reference material, and Table 3 is the data of tilapia collected in Dalseo Stream. The addition of references to Table 3 was not modified due to overlapping contents.

Point 15: Include reference values of each pure species attributes alongside the study fish Figure 3 (Line 202)

Response 15: No previous study raw data provided. The mean and standard deviation could not be obtained and could not be reflected in the Figure.

Point 16: Include more examples of pure populations Figure 4 for O. aureus and O. niloticus (native to Africa and Middle East) to distinguish from hybrids e.g. bold font for pure lineages, plain font hybrids. Include fish with known sampling location (line 222)

Response 16: Only objects identified as completely pure populations, including sampling regions, were included in this study analysis. Although it is registered as a pure population in other documents, there were many data that were not provided with sampling sites or were not pure populations. Therefore, containing multiple data could be confusing to the results, so I didn't add it.

Point 17: F3 instead of 3 etch, clearer that number relates to Fish Sample Number Table 4. Simplify Table no need to repeat numbers in Haplotype (Line 229)

Response 17: Each individual number is presented to show that the morphological and genetic analysis results were consistent.

Point 18: Colour code country origin (coloured circle) and highlight pure populations. Expand tree for study samples of O. aureus as well as O. niloticus. It is unclear the origin of the closest individual fish matching Haplotype 2 (Line 243)

Response 18: Revised. (Line 260)

Point 19: State ranges of earlier work (Line 264)

Response 19: Revised. (Line 284-285)

Point 20: Discuss why not O. mossambicus. Can this species be discounted by morphology? (Line 267)

Response 20: Unlike the two species (O. niloticus and O. aureus), O. mossambicus shows a clear difference in phenotype. Therefore, Mozambique tilapia was not considered in this study.

Point 21: The two species were ‘most closely associated with’ (Line 273). Speciation is not definitive as affiliations are to mixed populations of fish. Include fish for comparison where degree of hybridisation and purity is already known.

Response 21: It has been confirmed that Nile tilapia and Blutilapia live in Dalseocheon Stream. Among them, Nile tilapia was identified as a Filipino population as a result of the analysis of Figure 4 and 5, and hybridization is suspected.The issue of hybridization and purity will be covered in further research.

Point 22: Philippines likely hybrid lineage, provide evidence of relative species contribution and origin. What happens with morphological features in hybrids? Discuss likely % species contribution (Line 276)

Response 22: Because the difference in phenotypes for each individual was large, it was not possible to confirm the presence or absence of hybrid. Even in the morphological part, there were many overlapping parts with the shape of the original population, so it could not be distinguished.

Point 23: Reference and detail appropriate methods available eg SNPs. Include these characterised populations in the comparison (Line 283)

Response 23: Added examples of available methods. However, the detailed description of the method could not be added because the number of words in the journal was limited to 4000. (Line 304)

Point 24: List native fish present. Is there evidence of competition since the introduction of tilapia into the country? (Line 287)

Response 24: Currently, there is no ecological disturbance study in Korea. Failed to present native species due to the minimum number of words in the article type.

Point 25: Table S1 No need to provide sequence. State accession number, origin, whether pure or suspected introgressed and add Reference for comparative samples or state project sequences as ‘Current study’.

Response 25: We didn't modify it because I thought it was reasonable to present it as supplementary table.

Round 2

Author Response

Thank you for the helpful comments for improving the manuscript.

We substantially revised a previous version of our manuscript by considering all the comments and issues raised by the reviewer.

Point 1. although it was introduced earlier as one of the three previously mentioned species

it is probably observed for the first time in the nature as a feral population (Line 29)

Response 1: Yes, for the first time in this study, the habitat of natural populations was identified. No natural population had been reported before this study.

Point 2. in the lines 21-22 you said that: total of three species of tilapia (O. niloticus, O. mossambicus and O. aureus) were introduced to Korea before it is introduced but unrecorded? (Line 15-16)

Response 2: There is a record of being introduced and cultivated, but no natural population has been identified.

Point 3. it is not non-native everywhere in all aquatic systems (Line 52)

Response 3: It is not negative everywhere in aquatic systems, but it has a lot of influence as an non-native species.

Point 4. it sound strange (Line 110)

Response 4: Revised.

Point 5. it seems to be not just a lack of awareness but a lack of data on the distribution and species that are present (Line 309)

Response 5: Revised.

Point 6. it is introduced but unrecorded? line 22 (Line 316)

Response 6: There are reports of aquaculture, but there are no records of them inhabiting or living in natural populations.

Point 7. what kind of management measures are these (Line 330)

Response 7: Revised.

Reviewer 3 Report

Thank you for your reply.  Please find the following suggestions;

To reduce the word count and improve the wording delete words as highlighted -

Line 45, 68, 70, 74 etc

Line 67 indicate that (instead of found)

Dalseo Stream, with some introgression of O. niloticus with Oreochromis mossambicus suspected

Line 85 O. mossambicus is listed as being present in Korea.  Explain the decision in 2.3 Identification Line 265 to exclude this species in the comparison and why native population can be discounted.

Please add this statement and reference. 'Unlike the two species (O. niloticus and O. aureus), O. mossambicus shows a clear difference in phenotype by....... Therefore, Mozambique tilapia was not considered in this study (ref).'

Line 154, 261 Spacing

Line 178 Rephrase Kim and Park (1990) conducted karyotype analysis of farmed tilapia introduced in Korea and confirmed the presence of three species of On, Om and Oa.

Line 184 State methods used in this study [18]

Line 175 New sentence

Line 187 Replace with  - to

Line 188 barcoding using the cytochrome oxidase (COI) gene

Line 192 and inform on the management

Line 248 sentence case for Committee

Line 361 Rephrase from each of the 37 samples

Line 369 Philippines, Nigeria

Line 429 Rephrase.  registered in the NCBI were compared

Line 432 singular

Line 437 3.1 Photographic record

Line 438 re-phrase into sentence

Point 1 Please check the Journal recommendations of abbreviations.  Oreochromis in full first mention in Summary and Abstract as this may be published separately?

Line 446 indicated

Line 453 state how many

Line 489 number

Line 533 could not

Line 594 please be clear whether you are referring to farmed or native fish

Line 601 speculate introgression of O. nil to O. moss in Chinese and haplotype 1. 

Evidence that Philippines 'O.nil' are extensively introgressed with O.moss. https://doi.org/10.1111/j.1365-2109.1986.tb00111.x

This point of introgression has to be addressed in the interpretation of your results.  Identification of 'O.nil' is likely hybrid.

Line 603 provide references of these methods

Line 619 the fish 

Line 671 plural

Author Response

Thank you for the helpful comments for improving the manuscript.

We substantially revised a previous version of our manuscript by considering all the comments and issues raised by the reviewer.

Point 1. To reduce the word count and improve the wording delete words as highlighted - Line 45, 68, 70, 74 etc

Response 1: Revised.

Point 2. Line 67 indicate that (instead of found)

Response 2: Revised.

Point 3. Dalseo Stream, with some introgression of O. niloticus with Oreochromis mossambicus suspected

Response 3: We plan to cover it in further research.

Point 4. Line 85 O. mossambicus is listed as being present in Korea.  Explain the decision in 2.3 Identification Line 265 to exclude this species in the comparison and why native population can be discounted.

Response 4: Revised. Line 135-142, Table 1, Line 198-199

Point 4. Please add this statement and reference. 'Unlike the two species (O. niloticus and O. aureus), O. mossambicus shows a clear difference in phenotype by....... Therefore, Mozambique tilapia was not considered in this study (ref).'

Response 4: Revised. Line 135-142, Table 1, Line 198-199

Point 5. Line 154, 261 Spacing

Response 5: Revised.

Point 6. Line 178 Rephrase Kim and Park (1990) conducted karyotype analysis of farmed tilapia introduced in Korea and confirmed the presence of three species of On, Om and Oa.

Response 6: Revised.

Point 7. Line 184 State methods used in this study [18]

Response 7: It has been reported that the species was identified by phenotype.

Point 8. Line 175 New sentence

Response 8: Revised.

Point 9. Line 187 Replace with  - to

Response 9: Revised.

Point 10. Line 188 barcoding using the cytochrome oxidase (COI) gene

Response 10: Revised.

Point 11. Line 192 and inform on the management

Response 11: Revised.

Point 12. Line 361 Rephrase from each of the 37 samples

Response 12: Revised.

Point 13. Line 369 Philippines, Nigeria

Response 13: Revised.

Point 14. Line 429 Rephrase.  registered in the NCBI were compared

Response 14: Revised.

Point 15. Line 432 singular

Response 15: Revised.

Point 16. Line 437 3.1 Photographic record

Response 16: Revised.

Point 17. Point 1 Please check the Journal recommendations of abbreviations.  Oreochromis in full first mention in Summary and Abstract as this may be published separately?

Response 17: It's not against the rules of the Acronyms/Abbreviations/Initialisms.

Point 18. Line 446 indicated

Response 18: Revised.

Point 19. Line 453 state how many

Response 19: The object was not added because it was not significant. The individuals was not added because it was not important.

Point 20. Line 489 number

Response 20: Revised.

Point 21. Line 594 please be clear whether you are referring to farmed or native fish

Response 21: Revised.

Point 22. Evidence that Philippines 'O.nil' are extensively introgressed with O.moss. Point 1. https://doi.org/10.1111/j.1365-2109.1986.tb00111.x

Response 22: Revised.

Point 23. This point of introgression has to be addressed in the interpretation of your results.  Identification of 'O.nil' is likely hybrid.

Response 23: It's covered in the conclusion. Based on the results of this study, it is presumed to have been introduced in the Philippines.

Point 24. Line 603 provide references of these methods

Response 24: Revised.

Point 25. Line 619 the fish 

Response 25: Revised.

Point 26. Line 671 plural

Response 26: Revised.